# High Temperatures and Cardiovascular-Related Morbidity: A Scoping Review

**DOI:** 10.3390/ijerph191811243

**Published:** 2022-09-07

**Authors:** Kendra R. Cicci, Alana Maltby, Kristin K. Clemens, Ana Maria Vicedo-Cabrera, Anna C. Gunz, Éric Lavigne, Piotr Wilk

**Affiliations:** 1Department of Epidemiology and Biostatistics, Western University, London, ON N6G 2M1, Canada; 2Lawson Health Research Institute, London, ON N6C 2R5, Canada; 3Department of Medicine, Western University, London, ON N6A 5A5, Canada; 4ICES, London, ON N6A 5W9, Canada; 5St. Joseph’s Health Care, London, ON N6A 4V2, Canada; 6Institute of Social and Preventive Medicine, University of Bern, 3012 Bern, Switzerland; 7Oeschger Center for Climate Change Research, University of Bern, 3012 Bern, Switzerland; 8Department of Paediatrics, Western University, London, ON N6A 5W9, Canada; 9Child Health Research Institute, London, ON N6A 5W9, Canada; 10Air Health Science Division, Health Canada, Ottawa, ON K1A 0K9, Canada; 11School of Epidemiology and Public Health, Faculty of Medicine, University of Ottawa, Ottawa, ON K1G 5Z3, Canada

**Keywords:** high temperatures, cardiovascular, morbidity, extreme heat events, heat wave

## Abstract

The primary objective of this review was to synthesize studies assessing the relationships between high temperatures and cardiovascular disease (CVD)-related hospital encounters (i.e., emergency department (ED) visits or hospitalizations) in urban Canada and other comparable populations, and to identify areas for future research. Ovid MEDLINE, EMBASE, CINAHL, Cochrane Database of Systematic Reviews, and Scopus were searched between 6 April and 11 April 2020, and on 21 March 2021, to identify articles examining the relationship between high temperatures and CVD-related hospital encounters. Studies involving patients with pre-existing CVD were also included. English language studies from North America and Europe were included. Twenty-two articles were included in the review. Studies reported an inconsistent association between high temperatures and ischemic heart disease (IHD), heart failure, dysrhythmia, and some cerebrovascular-related hospital encounters. There was consistent evidence that high temperatures may be associated with increased ED visits and hospitalizations related to total CVD, hyper/hypotension, acute myocardial infarction (AMI), and ischemic stroke. Age, sex, and gender appear to modify high temperature-CVD morbidity relationships. Two studies examined the influence of pre-existing CVD on the relationship between high temperatures and morbidity. Pre-existing heart failure, AMI, and total CVD did not appear to affect the relationship, while evidence was inconsistent for pre-existing hypertension. There is inconsistent evidence that high temperatures are associated with CVD-related hospital encounters. Continued research on this topic is needed, particularly in the Canadian context and with a focus on individuals with pre-existing CVD.

## 1. Introduction

Average ambient temperatures in Canada have increased by 1.7 °C since 1948 and are expected to increase by an additional 1.8 °C by mid-century [1]. Extreme heat events and heat waves will become more frequent and intense, bringing prolonged periods of unusually high temperatures [2]. Exposure to high temperatures (i.e., high ambient temperature, extreme heat events, and heat waves) may precipitate heat stroke, heat edema, heat rash, heat stress, acute cardiovascular disease (CVD), and renal disease [3].

There have been many studies on the impact of high temperatures on CVD-related, respiratory-related, and all-cause mortality [4,5,6,7,8]. These studies suggest that mortality increases as temperatures move towards high extremes [9,10]. Certain populations are also disproportionately vulnerable to heat-related mortality, including the elderly [11,12,13], individuals with underlying medical conditions [14], those residing in urban settings [11,12], and those living at higher latitudes [9,11]. This is particularly concerning for Canada as it is not only located at a high latitude and has an aging population, but over 80% of Canadians reside in urban areas [15].

Less is known about the association between high temperatures and cardiovascular morbidity. A systematic review and meta-analysis on ambient temperature and cardiovascular hospitalizations from 2016 found that temperature was inconsistently associated with the risk of cardiovascular hospitalization [16]; however, the authors found that exposures to more extreme temperatures, such as those during heat waves, and over extended periods of time, were significantly associated with an increased risk of cardiovascular hospitalization, suggesting that the intensity and duration of exposure to high temperatures may influence CVD morbidity outcomes [16]. In Canada, CVDs were identified as the third and fifth most responsible diagnoses for hospitalization during the 2016–2017 fiscal year [17]. In 2020, total CVD costs (i.e., mortality, hospital costs, drugs, and long-term disability) were expected to reach CAD 28.3 billion [18]. However, there is a lack of clarity on the impact of high temperatures on those with established chronic health conditions. This is particularly important, as at least one in ten Canadian adults suffers from a chronic condition, with hypertension being the most prevalent [19].

Given that urban areas will see significant increases in the number of hot days (i.e., projections for 2051–2080 in Canada show that many cities could see at least four times as many days with temperature peaks above 30 °C per year on average) [20], there is a need to build upon previous reviews and update the available literature on the impacts of high temperatures on cardiovascular morbidity. Thus, the primary objective of this scoping review was to assess the relationship between high temperatures (e.g., heat waves, extreme heat events) and CVD-related hospital encounters, defined as CVD-related emergency department (ED) visits and hospitalizations, among urban populations. Relationships among populations with pre-existing CVDs were also examined. In the next sections, the methods will be introduced, the results presented, and the findings discussed, including the strengths and limitations of the review.

## 2. Methods

This scoping review was conducted according to the Preferred Reporting Items for Systematic reviews and Meta-Analyses extension for Scoping Reviews (PRISMA-ScR) checklist [21].

### 2.1. Search Strategy

Ovid MEDLINE, EMBASE, CINAHL, Cochrane Database of Systematic Reviews, and Scopus were searched between 6 April and 11 April 2020, and updated on 21 March 2021, to identify articles examining the association between high temperatures and CVD-related hospital encounters. A comprehensive search strategy for Ovid MEDLINE was developed by the research team and approved by a Research Librarian at Western University prior to database searching. The subject headings and Boolean operators were changed for use in subsequent databases (full Ovid MEDLINE search strategy is available by request).

### 2.2. Eligibility Criteria and Screening Process

The titles and abstracts of articles were screened by a single reviewer (KRC), who assessed eligibility using a priori inclusion and exclusion criteria. Studies that passed title and abstract screening underwent a full-text review to determine final inclusion status. The reference lists of all included articles were also searched for additional studies.

English language, peer-reviewed studies were included if they: (1) used high temperatures, extreme heat events, or heat waves as their exposure of interest; (2) examined any CVD-related hospital encounters as the primary outcome; and (3) reported a quantitative outcome (e.g., relative risk [RR], odds ratio [OR], or percentage of excess ED visits or hospitalizations). Studies examining the relationship between high temperatures and non-CVD-related hospital encounters among populations with pre-existing CVD were also included

Studies were limited to North America and Europe to produce results most generalizable to the Canadian context. Only studies conducted in urban settings were reported on, given the increased vulnerability of urban populations to high temperatures. Systematic reviews were included if at least half of included studies met the above criteria.

Primary studies were excluded if they were already part of an included systematic review (to avoid double counting). Editorials, unpublished manuscripts, dissertations, and abstracts were excluded. Studies that examined health outcomes based on climate change simulations (as opposed to empirical data) and those that did not examine ambient temperatures (i.e., heat therapy or heat experiments) were excluded. Grey literature was also excluded.

### 2.3. Data Extraction

Data were extracted by a single reviewer (KRC) according to pre-approved data charting fields. Key information included: (1) author(s) and year of publication, (2) study design, (3) study location and time period, (4) sample size, (5) exposure measure(s), (6) outcome(s), (7) risk factors and confounders, and (8) key findings. Supplemental materials were accessed for some studies to confirm results from the authors’ primary analyses.

### 2.4. Synthesis of Results

Once data were extracted, information on the association between high temperatures and total CVD-related hospital encounters, as well as for specific types of CVD hospital encounters (i.e., hypertension, ischemic heart disease (IHD), myocardial infarction, dysrhythmia, heart failure, cerebrovascular disease, and stroke), was synthesized and summarized.

## 3. Results

After deduplication, 483 unique articles (444 single studies and 39 reviews) were identified from the database searches (see Figure 1). Of these articles, 430 were excluded after title and abstract screening and 39 were excluded after full-text screening. Eight studies were added after scanning reference lists of included articles. The articles identified for inclusion in this review were published between 2008 and 2020.

Five systematic reviews and 17 single studies were included in the review. A flow diagram of inclusions and exclusions is presented in Figure 1.

### 3.1. Study Designs and Data Analysis

Five systematic reviews were included [16,22,23,24,25]. Four of the five systematic reviews also completed meta-analysis [16,23,24,25]. Of the remaining 17 studies, the majority utilized either a case-crossover design with conditional logistic regression [26,27,28], time series analysis utilizing Poisson regression [29,30,31,32,33,34], distributed lag nonlinear models [35], or generalized additive models [36]. Poisson regression and generalized linear regression [37,38], multiple stepwise regression analysis [39], canonical correlation analysis [40], and *t*-test [41,42] were also conducted. Detailed descriptions of the included studies can be found in Table 1.

### 3.2. Populations and Locations

The five systematic reviews included studies from North America, Europe, Asia, and Australia. Single studies were conducted in the United States [26,27,28,29,33,40,41], Canada [34,35,36,37], Southern Europe [31,38,39], Northern Europe [30,32], and Eastern Europe [42].

### 3.3. Exposures

Studies varied in how temperature exposure was defined and measured. Out of twenty-two studies, nine examined the effect of ambient air temperature, three used average ambient temperature [27,32,36], one maximum ambient temperature [37], and four minimum, average, and maximum ambient temperatures [22,24,25,38]. One study examined the effects of atmospheric temperature [39]. Additionally, two articles examined both ambient and apparent temperature [23,26], two studies examined the effects of ambient temperatures or heat waves [16,30], and one study examined the effects of both average ambient temperature and heat day exposure [35].

Eight of the twenty-two studies exclusively examined extreme temperature exposure. Four studies exclusively examined heat wave effects (i.e., when maximum temperatures exceed a predefined threshold for consecutive days) [29,40,41,42]; however, temperature thresholds ranged from the 90th to 99th percentile, while consecutive days needed at or above this threshold ranged from two to four. Three studies examined extreme heat events as defined by temperatures in the 99th percentile of temperature distribution for their respective locations (i.e., heat days) [31,33,34], and one study examined the effects of both heat waves and heat days (days exceeding the 95th percentile of temperature distribution) [28]. The distribution of study outcomes stratified by temperature exposure can be found in Table 2. In addition to differences in temperature exposure definitions, studies also varied in terms of how often temperatures were measured (e.g., hourly measurements versus daily measurements), and the length of lags examined between exposure and outcome (e.g., 0 days to a two-week period).

### 3.4. Risk Factors and Confounders

All primary studies accounted for individual-level risk factors and/or confounders in their analyses, with age, sex, and gender being the most common. The most common environmental confounders included air pollution [26,28,30,32,34,35,39], relative humidity [22,27,28,30,34,35,36,37,39], day of the week [26,27,29,30,31,32,33,35,41], and air pressure [30,36,38,39].

Studies differed in their methods of adjustment (i.e., study design and statistical analysis) for confounders and effect modifiers. For example, studies used multiple regression techniques to adjust for time-variant and time-invariant factors.

### 3.5. Outcomes

Eleven single studies and four systematic reviews assessed the impact of high temperatures on hospitalizations for CVD [16,23,24,25,27,28,30,31,33,35,36,37,38,40,42]. Five single studies investigated the relationship between high temperatures and ED visits [29,32,34,39,41], with the remaining primary study and systematic review examining both hospitalizations and ED visits [22,26].

Eight articles reported the impact of high temperatures on hospital encounters for a specific cardiovascular disease [22,24,27,28,36,37,38,42]; two studies exclusively focused on hospital encounters for total CVD [16,39]; and ten studies reported hospital encounters for multiple cardiovascular diseases [23,25,29,30,31,32,33,35,40,41]. Lastly, two studies examined whether cardiovascular comorbidities increased the risk for heat stress and dehydration-related ED visits [26] or CVD and respiratory-related ED visits [34].

Due to the heterogeneity of reported outcomes, the results of studies are presented descriptively: total CVD, hyper/hypotension, ischemic heart disease, myocardial infarction, dysrhythmia, heart failure, cerebrovascular disease, and stroke. The distribution of effect estimates can be found in Table 3.

Cardiovascular Disease (Total). Thirteen studies examined the effects of high temperatures on total CVD hospital encounters. Four primary studies and one systematic review and meta-analysis reported a positive relationship between high temperatures and total CVD-related ED visits [29,32,41] and hospitalizations [23,40]. Of these, four articles reported statistically significant results [29,32,40,41] and two reported nonsignificant results [23,29]. In contrast, one systematic review and meta-analysis and three single studies reported decreases in CVD hospitalizations [25,30,31] and CVD-related ED visits [39]; however, only two studies reported statistically significant results [31,39]. In addition, one systematic review and meta-analysis and two single studies reported both increases and decreases in the risk of CVD and circulatory disease hospitalizations due to high temperatures. Results differed based on exposure definition [16], lagged effects [26], and age [33].

Only one study examined whether underlying cardiac diseases modified the relationship between high temperatures and respiratory-related ED visits. No association was reported between comorbid cardiac diseases and ED visits during heat days (i.e., temperature exceeding 28.4 °C) [34].

Hypertension/hypotension. Six studies examined the association between heat waves or heat days and hospital encounters related to hyper/hypotension. Four of the six studies reported increases in the risk of ED visits [29,41] or hospitalizations [35,40]. Of the six studies, two studies reported statistically significant results [29,41], while four reported nonsignificant increases [29,35,40,41]. It is important to note that Chen et al. and Fuhrmann et al. assessed effects using either various definitions of heat waves [29] or multiple heat waves [41], resulting in both statistically significant and nonsignificant results.

Only Lavigne et al. [34] and Adeyeye et al. [26] reported associations between underlying hypertension and heat-related hospital encounters. Lavigne et al. [34] concluded that individuals with underlying hypertension did not have a statistically significant higher risk of CVD- or respiratory-related ED visits on heat days (28.4 °C or higher), although the direction of the effect was positive. Adeyeye et al. [26] reported that individuals with hypertension had a slightly higher risk of visiting the ED than those without hypertension during high ambient temperature exposure.

Ischemic Heart Disease. Seven studies examined the effects of high temperatures on hospital encounters for ischemic heart disease (IHD). Two studies reported statistically significant increases in the risk of IHD ED visits [41] or IHD hospitalizations [40] during heat wave exposure; however, three studies reported both increases and decreases in the risk of IHD-related hospital encounters [29,30,33] and one study reported no change in IHD hospital admissions during heat wave exposure [42].

Bayentin et al. [36] examined the association between high ambient temperatures and hospitalizations for IHD, reporting that women aged 45 to 64 years were more likely to experience an increased risk of IHD compared to men of the same age; however, no other significant patterns emerged from the analysis.

Acute Myocardial Infarction. Six studies and four systematic reviews examined the effect of high temperatures on acute myocardial infarction (AMI) hospital encounters, with the majority reporting a positive relationship. Two systematic reviews and four single studies reported increases in the risk of AMI ED visits [41] or hospitalizations [22,24,25,27,40] at high temperatures. Of these, three reached statistical significance [24,27,40] and three reported nonsignificant results [25,27,41]. The review by Bhaskaran et al. [22] reported that seven of thirteen studies had statistically significant results but did not attempt to pool effect estimates to compute overall significance.

Only one systematic review and meta-analysis reported a statistically nonsignificant decrease in the risk of AMI hospitalizations during exposure to high ambient temperatures [23]; however, two single studies reported both increases and decreases in IHD hospitalizations during high-temperature exposures [30,37]. Results were dependent on heat wave definition [30] and gender [37]. Adeyeye et al. [26] found no association between underlying AMI conditions and ED visits for heat stress or dehydration due to high temperatures.

Dysrhythmia. Five studies examined the association between high temperatures and hospital encounters for cardiac dysrhythmias/arrhythmias. Only two studies exclusively reported increases in hospitalizations [40] or ED visits [41] for cardiac dysrhythmias during heat waves; however, one study did not reach statistical significance. Conversely, three studies reported both increases and decreases in the risk of heart failure morbidity [29,30,35]. Results were dependent on exposure definition [29,30,35], age [30], and immediate versus lagged effects [29].

Heart Failure. Three studies examined the association between high temperatures and hospital encounters for heart failure. Fuhrmann et al. [41] reported nonsignificant increases in heart failure ED visits across three heat waves included in the study period. However, Chen et al. [29] reported that congestive heart failure risk significantly increased in heat waves defined by minimum temperature. Adeyeye et al. [26] reported that individuals without pre-existing heart failure had a statistically significant increased risk of visiting the ED for heat stress, whereas the increased risk among individuals with pre-existing heart failure did not reach statistical significance.

Cerebrovascular Diseases/Stroke. Eleven studies examined the association between high temperatures and hospital encounters for stroke. Two systematic reviews and meta-analyses and five single studies reported outcomes for total cerebrovascular-related hospital encounters. Both meta-analyses reported nonsignificant decreases in risk for total stroke during exposure to high temperatures [23,25], with only one single study exclusively reporting a statistically significant increase in risk [32]. The remaining four studies reported both increases and decreases in the risk of total stroke [28,30,31,33], with results differing based on age [28,30,31,33], exposure definition [30], and immediate versus lagged effects [28].

When examining only ischemic stroke-related hospital encounters, one systematic review and three single studies reported increases in ischemic stroke hospitalizations [23,40] or ED visits [29,41]. Of these, two reached statistical significance [29,40], while three reported nonsignificant increases [23,29,41]. Only one study exclusively reported a decreased risk of ischemic stroke hospitalizations [38] with the remaining study [28] reporting both increases and decreases in risk depending on exposure definition and immediate versus lagged effects.

One systematic review and three single studies examined the effects of high temperatures on the risk of ED visits [41] or hospitalizations [23,28,38] related to hemorrhagic strokes. The systematic review and meta-analysis reported a nonsignificant decreased risk of hemorrhagic stroke hospitalization [23], while one study exclusively reported a nonsignificant increase [38]. Conversely, two single studies reported both nonsignificant increases and decreases in risk [28,41], with results differing based on exposure [28,41] and immediate versus lagged effects [28].

## 4. Discussion

### 4.1. Main Findings

In this scoping review of 22 studies, the findings suggest that hospital encounters for total CVD, AMI, hyper/hypotension, and ischemic stroke may be influenced by high temperatures. Total CVD was the most studied outcome and several papers found positive relationships between high temperatures and total CVD-related ED visits [29,32,41] and hospitalizations [23,40]. Studies that reported both increases and decreases in the risk of CVD and circulatory disease hospitalizations due to high temperatures noted that these differences are primarily due to exposure definition [16], lagged effects [26], and age [33]. The relationship between high temperatures and IHD-, heart failure-, dysrhythmia-, and other stroke-related hospital encounters, however, was inconsistent. Age was identified as a possible risk factor, as those over 65 years of age had a higher risk of ED visits or hospitalizations for total CVD, IHD, AMI, total stroke, ischemic stroke, heart failure, and dysrhythmia [28,30,31,33,41], demonstrating agreement with previously published studies [4,43,44]. This may be due to physiological changes caused by aging; older individuals experience central cardiovascular strain in addition to reduced capacity to thermoregulate and redistribute blood flow thereby increasing their vulnerability to high temperatures [43]. Older individuals may also have lower recall rates of heat warnings and confusion related to heat-health risks, which can result in a decreased use of adaptive behaviours [45]. Additionally, older individuals may experience less physical independence and reduced capacity to adapt their activities to changes in temperatures compared to younger individuals [45]. It is important to note that two studies from Quebec, Canada, reported an increased risk of IHD and STEMI hospital admissions among younger age groups, particularly women 45–64 years old and 55 years or younger, respectively [36,37]. This difference may be due to non-adjustment for confounding factors. For example, Bayentin et al. [36] noted that the geographical regions with statistically significant higher risks of STEMI admissions also displayed higher material deprivation and higher prevalence of smoking, particularly among those in the 45–64-year-old age group. Thus, the effects of high temperatures on total CVD and specific cardiovascular conditions may vary by age due to factors beyond physiological ones, including sociodemographic and environmental conditions.

Sex/gender may also influence the association between high temperatures and hospital encounters for IHD, cerebrovascular events, and AMI. Bayentin et al. [36] and Gebhard et al. [37] found that hospitalizations for IHD and STEMI were higher among younger women compared to men of the same age. AMI has been reported to be the first indication of IHD in approximately 50% of patients in a previous cross-sectional study [46]. Women have been found to have a higher core temperature, skin temperature, heart rate, and blood pressure compared to men, which can decrease tolerance to heat; however, studies that match male and female participants on body size and fat percentage have shown minimal differences in thermoregulation, indicating that physical characteristics may be more important than sex when determining heat tolerance [47]. Ha et al. [28] reported that men were significantly more likely to be hospitalized for stroke than women. These results are consistent with results from a systematic review on stroke epidemiology that found that men had a 33% higher stroke incidence rate and a 41% higher stroke prevalence rate than women, possibly due to a protective effect of estrogens against ischemic stroke, or the higher prevalence of smoking and hypertension among males [48]. Sociodemographic differences between genders such as social contact/isolation might also explain differences. For example, during a heat wave in Paris, heat risk increased for unmarried men but not for unmarried women [14]. In a systematic review on heat-related illness in men and women, the authors found that women are at reduced risk of heat illness than men across all ages, severity of heat illness, and occupations; however, the reason for these differences was not clear, and it was postulated to be due to sex-related behavioural differences [49]. It is likely that the intersectionality between individual and environmental factors affects the vulnerability and adaptability of some populations and thus additional research in the area of CVD morbidity that stratifies the results for sex and gender is needed. Such research will provide a more thorough understanding of how heat affects genders differently which is needed for developing adaptation and mitigation strategies.

It is not surprising that intensity and duration of exposure to high temperatures may also influence CVD-related hospital encounters. This was particularly notable when examining the effects of high temperatures on total CVD hospital encounters. Studies that used ambient, apparent, or atmospheric temperature more often reported statistically significant decreases in risk as temperatures increased; however, studies that used extreme temperature exposures, such as heat waves, were more likely to report statistically significant increases in risk. Examining exposure to heat waves, only one of seven studies reported a statistically significant decreased risk of any outcomes [30], whereas four of twelve studies examining ambient, apparent, or atmospheric temperatures reported statistically significant decreases in risk [26,27,37,39]. These findings are consistent with current understandings of temperature–morbidity and temperature–mortality relationships, whereby a U- or V-shaped exposure–outcome relationship has been reported. That is, morbidity and mortality continue to increase beyond both a high and low-temperature threshold [9,10,27,50]. Chen et al. [29] consistently reported higher risk when heat waves were defined by minimum rather than maximum temperature, which may seem to contrast with previous findings; however, the reason for this discrepancy may have been due to the higher number of heat waves defined by minimum temperature included in the study, leading to a higher power to detect a significant effect. This may be due to the greater influence that urban heat island effects have on minimum, compared to maximum, temperature [51]. Additionally, the authors noted that there was only moderate concordance across heat wave definitions; therefore, heat waves defined by minimum temperature may capture different aspects of extreme heat, and subsequently different outcomes than heat waves defined by the maximum temperature [29].

Only two studies examined how pre-existing CVD influenced the relationship between high temperatures and non-CVD hospital encounters, with inconsistent results across CVD subtypes. No association was reported for pre-existing heart failure, AMI, or total CVD, and conflicting results were reported for pre-existing hypertension, depending on the exposure (extreme heat versus high ambient temperature) and outcome (CVD or respiratory-related ED visits versus heat stress or dehydration-related ED visits). It is unknown if the presence of pre-existing CVD increases vulnerability to high temperatures, or whether individuals with pre-existing CVD exhibit adaptive behaviours, such as staying indoors or limiting strenuous physical activity to minimize their risk. Additionally, the comorbid nature of many of these cardiovascular conditions makes it difficult to disentangle the potential for their independent effects on outcomes.

### 4.2. Strengths and Limitations

This scoping review has some noteworthy strengths. The PRISMA-ScR checklist was followed, and a search protocol was developed in collaboration with the research team and approved prior to initiating this review to ensure transparency and rigour in the methodology and to aid in replicability in the future. Additionally, the search strategy was developed in collaboration with a research librarian at Western University, resulting in a thorough search that spanned five research databases and included reference list scanning to help minimize publication bias. Furthermore, although the results presented for each subtype of CVD were based on a small number of studies, they reflect the current state of research on the relationship between high temperatures and CVD-related ED visits and hospitalizations, indicating a need for future research on this topic, particularly in the Canadian urban context.

This scoping review is not without its limitations. One reviewer completed the title/abstract, full-text, and reference list screening. This constraint may have introduced reviewer bias, potentially excluding relevant studies from the final analysis. Publication bias is also possible due to the absence of grey literature searching and excluding any non-English publications. Only ED visits and hospitalizations related to CVD were included; therefore, the high temperatures–CVD morbidity relationship outside of these settings were not examined and reported.

In addition, this review is limited by the heterogeneity of included studies, making synthesis of the results and generalizability to the Canadian context difficult to ascertain. Studies varied in terms of their population size and composition, the confounders and risk factors included in the analysis, exposure measurements, the geographical areas and climate regions examined, and whether lag effects were accounted for. It is likely that features of the natural, built, and social environments unique to each neighbourhood may modify the effects of high temperatures on health in different geographic areas. Furthermore, our review focused on urban areas, and there is a need to further study the effect of high temperatures on cardiovascular health in rural areas of Canada. A number of studies also reported statistically nonsignificant or inconsistent results. There are various reasons why studies may produce conflicting results, such as conducting analysis on a low number of identified cases, using a small sample size, or failing to control for relevant confounders. Therefore, solid conclusions about the associations reported across reviewed studies were unable to be derived. Moreover, there are no apparent differences in the nature of the results, based on the year of publication. Lastly, the scientific qualities of the studies included were not critiqued, a process typically completed in systematic reviews. The aim of this scoping review was to identify the state of the knowledge on CVD-related ED visits and hospitalizations, which can be used to inform future primary studies or systematic reviews.

### 4.3. Research Gaps

Only four studies examining the relationship between high temperatures and CVD-related hospital encounters conducted in Canada were identified in this review, highlighting the need for Canadian-specific research. As temperatures in Canada continue to increase at a rate greater than the global average, and extreme weather events including heat waves are projected to increase in intensity and severity in the years to come, understanding our vulnerability to such events is crucial for future adaptation planning. Additionally, only two studies were retrieved that examined the effects of high temperatures on populations with pre-existing CVD or CVD risk factors. Research should focus on understanding the potential modifying influence of pre-existing CVD on the association between high temperatures and hospital encounters in the future, as this population has the potential for increased vulnerability to increasing temperatures. Although total CVD, AMI, and cerebrovascular disease/stroke had adequate representation in this review, other subtypes of CVD have been understudied, such as dysrhythmia and heart failure. Studies examining high-temperature effects on specific subtypes, as opposed to total CVD, are warranted in the future, especially given recent findings that some subtypes are more affected by high temperatures than others. Furthermore, examining other definitions of CVD morbidity, beyond ED visits and hospitalizations, and including studies from around the world may provide a better understanding of the relationship between high temperatures and CVD morbidity and possibly reduce the heterogeneity of the findings. Finally, there is a need to explore the causal mechanism linking high temperatures with CVD, and as such, study designs should include analysis to assess the moderating and mediating effects of some individual- and area-level factors.

## 5. Conclusions

High temperatures may be associated with increased hospital encounters related to total CVD, hyper/hypotension, acute myocardial infarction (AMI), and ischemic stroke. Age, sex, and gender, and the intensity/duration of exposure to high temperatures (e.g., heat wave exposure versus heat day or high ambient temperature exposure) may also modify the relationship between high temperatures and various CVD-related hospital encounters; however, pre-existing CVD subtypes either showed no effect or inconsistent effects on the association between high temperatures and CVD-related ED visits or hospitalizations.

## Figures and Tables

**Figure 1 ijerph-19-11243-f001:**
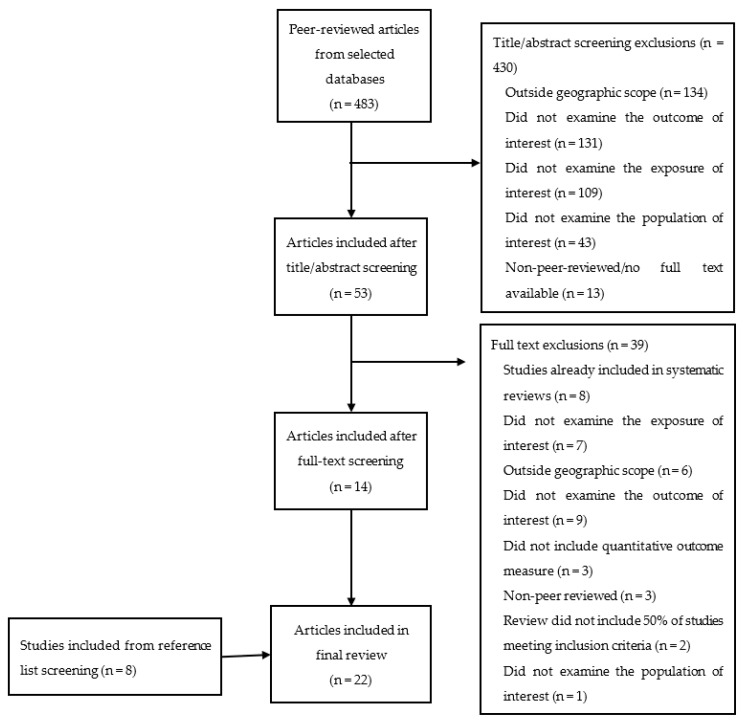
Prisma-ScR Flow diagram for included and excluded studies.

**Table 1 ijerph-19-11243-t001:** Data extraction from included studies.

Study	Study Design	Location and Study Period	Sample Size	Exposure	Outcome	Risk Factors and Confounders	Key Findings
Adeyeye et al. [26]	Case crossover	EDs and hospitals in New York State, USA Summer months 2008–2012	n_heat stress_ = 8703 n_dehydration_ = 59,828 n_CVD_ = 827,051	Ambient temperature (maximum, heat index) Measured daily	Risk Ratio CVD, heat stress, and dehydration	Risk factors/effect modifiers: urbanicity, race, ethnicity, pre-existing chronic illnesses, ozone, PM_2.5_, the month of exposure Confounders: individual time-invariant factors, day of the week, season	-For every 1 °C increase in maximum temperature, the risk for ED visits statistically significantly increased after a lag of 4–7 days. -Presence of pre-existing CVD was not associated with high temperatures and heat stress/dehydration.
Bai et al. [35]	Time series	Hospitals in Ontario, Canada 1996–2013	n_hypertension_ = 50,788; n_arrhytmia_ = 345,052	Ambient temperature (average: mild heat between optimum temperature and 97.5th percentile) and extreme heat (higher than the 97.5th percentile). Measured hourly	Cumulative relative risk; attributable fraction Hypertension, arrhythmia	Risk factors/effect modifiers: age, sex, comorbid conditions, intake of anti-hypertensive medications Confounders: NO_2_, O_3_, humidity, influenza, holiday effects, day of the week	-High temperatures were not statistically significantly associated with hypertension or arrythmia hospitalizations in Ontario.
Bayentin et al. [36]	Time series	Hospitals in Quebec, Canada 1989–2006	15 health regions	Ambient temperature (average) Measured daily	Percent change IHD	Risk factors/effect modifiers: age, sex, health region, dew-point temperature Confounders: relative humidity, precipitation, atmospheric pressure, ground snow, time trends, day of the season	-Hot episodes during summer months were associated with increases in IHD hospital admissions but also showed decreased risks in particular regions. -Differences in risk were present for women aged 45–64 compared to men of the same age.
Bhaskaran et al. [22]	Systematic review	Hospitals and EDs in North America, Europe, Asia, Australia Searched database inception-2009	19 studies	Ambient temperature (minimum, maximum, or average)	Narrative synthesis MI	Assessed studies for adjustment for air pollution and other potential confounders	-Both hot and cold weather appeared to have detrimental effects on the short-term risk of MI.
Bunker et al. [23]	Systematic review and meta-analysis	Hospitals in North America, South America, Europe, Asia, Australia Searched databases 1 January 1975–24 July 2015	n_cerebrovasular_ = 8 studies included in meta-analysis n_cardiovascular_ = 11 studies included in meta-analysis	Ambient temperature, apparent temperature, diurnal temperature range	Percent change in risk Total cerebrovascular, ischemic stroke, haemorrhage, total CVD, MI	Risk factors/effect modifiers: age Confounders: lag days, temperature variables, type of hospital admission, outcome classification	-Ambient temperature was not found to statistically significantly increase the risk for cerebrovascular or cardiovascular morbidity -There was a significant association between ambient temperature and CVD and cerebrovascular mortality.
Chen et al. [29]	Time series	EDs in Atlanta metropolitan area, USA Warm season between 1993 and 2012	N = 9,856,015	Heat waves (temperatures exceeding 98th percentile for 2 or more consecutive days) Measured daily	Relative risk Total circulatory, hypertension, IHD, dysrhythmia, congestive heart failure, ischemic stroke	Risk factors/effect modifiers: continuous air temperature, dew-point temperature, CVD subtype Confounders: day of the week, holidays, and time trends	-Minimum and maximum temperature during heat waves can increase the morbidity risk of specific CVDs. -Minimum temperature-defined heat waves were associated with statistically significant increases in ED visits for total circulatory disease, hypertension, IHD, dysrhythmia, and ischemic stroke. -Maximum temperature-defined heat waves were statistically significantly associated with increases in hypertension and ischemic stroke. -Some CVDs are associated with immediate changes in morbidity, while others displayed a one-day lag.
Fuhrmann et al. [41]	*t*-test	EDs in North Carolina, USA Warm season between 2007 and 2011	N = 100 counties in North Carolina	Heat waves (at least one heat product issued and verified across four or more NWS county warning areas for five or more consecutive days) Measured daily	Percent change Total CVD, IHD, AMI, dysrhythmia, heart failure, hemorrhagic stroke, ischemic stroke, aneurysm, hypotension	Confounders: day of the week, day in season	-The number of ED visits for total CVD was significantly elevated during all three heat waves; however, examining specific subtypes of CVD, only the risk of hypotension and IHD statistically significantly increased, and only during specific heat waves.
Gebhard et al. [37]	Single centre retrospective study	Montreal Heart Institute in Montreal, Canada 2010–2014	N = 2199	Ambient temperature (maximal, >15 °C); season; daylight (>12 h) Measured daily	Relative risk STEMI	Risk factors/effect modifiers: cardiovascular risk factors, age, sex Confounders: total rain, total snowfall, relative humidity, wind speed	-The effect of high temperatures on STEMI was age and sex-dependent. High temperatures may be associated with an increased risk of STEMI among young women less than 55 years, compared to older women or men.
Guirguis et al. [40]	Observational study using canonical correlation	Acute care facilities in California, USA Warm season between 1999 and 2009	N = 6 California subregions	Heat waves (periods where there is a significant correlation between temperature and health outcomes in addition to strong temperature and health anomalies) Measured daily	Percent change Total CVD, IHD, AMI, dysrhythmia, hypertension, ischemic stroke	Risk factors/effect modifiers: age, race/ethnicity Confounders: viral activity, weekends and holidays, long-term trends	-On average, hospital admissions increased 7% on peak heat wave days, with statistically significant increases in admissions for total CVD and all subtypes excluding essential hypertension.
Ha et al. [28]	Case crossover	Hospitals in Allegheny County, USA Warm season between 1994 and 2000	N = 12,195	Extreme heat (temperature greater than the 95th percentile); heat waves (two or more consecutive heat days) Measured daily	Odds ratio Total stroke, ischemic stroke, hemorrhagic stroke	Risk factors/effect modifiers: type of stroke; gender; age Confounders: time-invariant factors, air pollution, relative humidity	-Heat waves had a greater effect on stroke hospitalizations than heat days, demonstrating a lag effect for total stroke and ischemic stroke. -Hemorrhagic stroke does not appear to be affected by high temperatures. -Men and those 80 years and older may be more susceptible to stroke hospitalization during heat days or heat waves than the general population.
Hotz and Hajat [32]	Time series	EDs in Greater London, UK April 2007–March 2012	n = 13,400,000	Ambient temperature (average) Measured daily	Percent change Total CVD, cerebrovascular disease	Risk factors/effect modifiers: deprivation levels, age Confounders: influenza, PM_10_, ozone, day of the week, public holidays, humidity	-There was a small but statistically significant increase in cardiac and cerebrovascular ED visits associated with a 1 °C increase in daily average temperatures.
Isaksen et al. [33]	Time series	King County hospitals in Washington, USA Warm season between 1990 and 2010	N = 752,151	Extreme heat (day with the temperature reaching the 99th percentile) Measured daily	Relative risk/percent change Circulatory disease, total CVD, IHD, cerebrovascular disease	Risk factors/effect modifiers: age, gender, nighttime temperature Confounders: SES, synoptic weather type, admission source, admission type	-Heat, expressed as humidex, was statistically significantly associated with increased hospital admissions for circulatory disease and CVD, but only among those 85 years or older. CVD admissions also significantly decreased among those 45–64 years old. -No significant association between high temperatures and IHD or cerebrovascular disease for age groups studied.
Jimenez-Conde et al. [38]	Observational study	Hospital del Mar in Barcelona, Spain 2001–2003	N = 1286	Ambient temperature (minimum, average, maximum) Measured daily	Relative risk All strokes, non-lacunar strokes, lacunar strokes, intracerebral hemorrhage	Risk factors/effect modifiers: stroke subtype Confounders: season, air pressure variations, daily meteorological variables	-Temperature change did not have a significant effect on stroke incidence after controlling for meteorological variables. -Most variations in incidence could be explained by variations in air pressure as opposed to temperature.
Lavigne et al. [34]	Time series	EDs in Toronto, Canada 2002–2010	n_respiratory_ = 562,738 n_CVD_ = 292,666	Extreme temperature (day with the temperature reaching the 99th percentile) Measured daily	Relative risk CVD, respiratory disease	Risk factors/effect modifiers: comorbid health conditions Confounders: influenza, air pollution, relative humidity, seasonal effects, sub-seasonal cycles, and long-term trends	-Underlying hypertension or cardiac diseases did not significantly increase the risk of CVD or respiratory ED visits during extreme temperatures; however, underlying diabetes significantly increased the risk of CVD ED visits.
Martínez-Solanas and Basagaña [31]	Time series	Hospitals in Spain 1997–2013	n_cardiovascular_ = 4,475,984 n_cerebrovascular_ = 1,320,418	Extreme temperature (day with the temperature reaching the 99th percentile) Measured daily	Percent change Total CVD, cerebrovascular disease	Risk factors/effect modifiers: age, sex Confounders: seasonality, day of the week, holidays, influenza	-Heat days were associated with a statistically significant decrease in CVD hospitalizations. -Cerebrovascular hospitalizations increased for most age groups; however, were not statistically significant.
Phung et al. [16]	Systematic review and meta-analysis	Hospitals in North America, Europe, Australia, Asia Does not state the publication range searched for eligible studies	N = 64 studies	Ambient temperature/heat wave/cold spell	Pooled relative risk Total CVD	Confounding included in the quality appraisal of studies	-There was a statistically significant increase in CVD hospitalizations during a heat wave; however, these findings did not extend to heat exposure (i.e., high temperatures that do not reach heat wave intensity or duration).
Pintaric et al. [39]	Observational study	University hospital EDs in Zagreb, Croatia 2008–2010	N = 20,228 at 2 sites	Atmospheric temperature (average) Measured daily	Spearman’s rank correlation Total CVD	Confounders: air pollution, seasons, atmospheric pressure, relative humidity	-Temperature was statistically significantly negatively correlated with CVD-related ED visits.
Rowland et al. [27]	Time stratified case-crossover	Hospitals in New York State, USA 2000–2015	n = 791,695 admissions	Ambient temperature (average) Measured hourly	Percent change AMI	Risk factors/effect modifiers: age, sex, time of day, season, relative humidity, first/recurrent status Confounders: residential ZIP code, year, month, day of the week, hour of day	-Increases in hourly ambient temperature may be associated with an increased risk of MI among men and those experiencing first MI. -Temperature demonstrated a decreasing risk as lag time increased. -While high temperatures may be associated with an increased risk of MI in the afternoon, evening, and night, it had a protective effect against MIs occurring in the morning.
Sohail et al. [30]	Time series	Hospitals in Helsinki, Finland June 2001–August 2017	Does not state	Ambient temperature (average), heat wave (four or more consecutive days with the temperature reaching the 90th percentile or three or more consecutive days with the temperature reaching the 95th percentile) Measured daily	Percent change Total CVD, AMI, IHD, cerebrovascular disease, arrhythmia	Risk factors/effect modifiers: age Confounders: NO_2_, O_3_, PM_10_, PM_2.5_, relative humidity, barometric pressure, pollen	-Ambient temperature was not significantly associated with any changes in disease outcomes; however, heat waves appeared to be statistically significantly associated with an increased risk of AMI among those 65–74 years and a decreased risk of arrhythmia and cerebrovascular disease hospitalizations among those less than 75 years and 18–64 years, respectively.
Sun et al. [24]	Systematic review and meta-analysis	Hospitals in North America, Europe, Asia Search conducted 31 August 2017, and included studies from database inception to search date	N = 13 studies included in the meta-analysis for heat exposure	Ambient temperature (minimum, average, maximum)	Pooled relative risk AMI	Risk factors/effect modifiers: temperature, latitude, lag days Confounding included in the quality appraisal of studies	-Increasing ambient temperature was found to be statistically significantly associated with AMI hospitalizations. -A one degree increase in latitude was statistically significantly associated with a decreased risk of AMI during exposure to high ambient temperature.
Turner et al. [25]	Systematic review and meta-analysis	Hospitals in North America, Europe, Australia, Asia Search conducted October 2010–January 2012 and included studies from database inception to current search	N = 21 studies	Ambient temperature	Pooled relative risk Total CVD, stroke, acute coronary syndrome/MI	Risk factors/effect modifiers: lagged effects, latitude Confounding included in the quality appraisal of studies	-No apparent association was found between increases in ambient temperature and cardiovascular morbidity.
Zacharias et al. [42]	*t*-test	Hospitals in 19 regions in Germany 2001–2010	N = 14,959,190	Heat wave (three or more consecutive days with the temperature exceeding the 97.5th percentile) Measured hourly	Percent change IHD	Risk factors/effect modifiers: gender, subgroups of ischemic diseases, geographic region Confounders: long-term trends, seasonal fluctuations	-Heat waves had no observed influence on hospital admissions due to IHD, although authors did report a significant increase in IHD mortality.

Abbreviations: AMI, acute myocardial infarction; CVD, cardiovascular disease; ED, emergency department; IHD, ischemic heart disease; MI, myocardial infarction; STEMI, S-T-elevation myocardial infarction.

**Table 2 ijerph-19-11243-t002:** Distribution of study exposures and examined outcomes.

Exposures	Total CVD	Hypertension	Hypotension	IHD	AMI	Dysrhythmia	Heart Failure	Cerebrovascular Disease	Heat Stress and Dehydration	Total Respiratory
Ambient temperature	6	1	0	2	6	1	0	5	1	0
Apparent temperature	2	0	0	0	1	0	0	1	0	0
Atmospheric temperature	1	0	0	0	0	0	0	0	0	0
Heat wave	5	2	2	5	3	4	2	5	0	0
Heat day	3	1	0	1	0	1	0	3	0	1

Note: A study may include more than one cardiovascular-related outcome or pre-existing condition. Abbreviations: AMI, Acute myocardial infarction; CVD, cardiovascular disease; IHD, Ischemic heart disease.

**Table 3 ijerph-19-11243-t003:** Statistical significance and direction of effect between temperature exposures and outcomes.

High-Temperature Exposure	Cardiovascular Disease	Statistical Significance			
		Significant (+)	Nonsignificant (+)	Significant (−)	Nonsignificant (−)
Ambient temperature	Total CVD	Adeyeye et al. [26]: lag4,6,7 Hotz and Hajat [32]	Bunker et al. [23] Sohail et al. [30]: 65–74	Adeyeye et al. [26]: lag0–1	Phung et al. [16] Sohail et al. [30]: 18–64, 75+ Turner et al. [25]
	Hyper/hypotension		Adeyeye et al. [26] ^a^ Bai et al. [35]		
	Ischemic Heart Disease	Bayentin et al. [36]: women 45–64 compared to men 45–64			Sohail et al. [30]
	Acute Myocardial Infarction	Bhaskaran et al. [22]: 7/13 studies Gebhard et al. [37]: women ≤ 55 years Rowland et al. [27]: lag0–5 (males, all ages, afternoon, evening, night, first-time AMI) Sun et al. [24]	Rowland et al. [27]: lag0–5 (females), lag0–47 (females, males, all ages, night MI, first MI) Sohail et al. [30]: 65–74, 75+ Turner et al. [25]	Gebhard et al. [37]: men ≤ 55 years during temperatures > 15 °C or daylight > 12 h Rowland et al. [27]: lag0–5 and lag0–47 (morning MI)	Adeyeye et al. [26] ^a^ Bunker et al. [23] Gebhard et al. [37]: men ≤ 55 years during spring/summer Rowland et al. [27]: lag0–5 (recurrent MI), lag0–47 (afternoon MI, evening MI, recurrent MI) Sohail et al. [30]: 18–64 years
	Dysrhythmia		Sohail et al. [30]: 65–74		Sohail et al. [30]: 18–64, 75+
	Heart Failure		Adeyeye et al. [26] ^a^		
	Total Stroke	Hotz and Hajat [32]			Bunker et al. [23] Sohail et al. [30] Turner et al. [25]
	Ischemic Stroke		Bunker et al. [23]		Jimenez-Conde et al. [38]
	Hemorrhagic Stroke		Jimenez-Conde et al. [38]		Bunker et al. [23]
Apparent temperature	Total CVD	Adeyeye et al. [26]: lag4,6,7	Bunker et al. [23]	Adeyeye et al. [26]: lag0–1	Adeyeye et al. [26]: lag2
	Acute Myocardial Infarction				Bunker et al. [23]
	Total Stroke				Bunker et al. [23]
	Ischemic Stroke		Bunker et al. [23]		
	Hemorrhagic Stroke				Bunker et al. [23]
Atmospheric temperature	Total CVD			Pintaric et al. [39]	
Heat day	Total CVD	Isaksen et al. [33]: percent change in hospitalizations among those 85 years+, circulatory and cardiovascular	Lavigne et al. [34] ^b^: lag0–1 respiratory ED visit	Isaksen et al. [33]: percent change in hospitalizations among those 45–64 years, cardiovascular Martínez-Solanas and Basagaña [31]: 16–84 years	Isaksen et al. [33]: RR estimate, the percent change in hospitalizations among those 45–84 years, circulatory Lavigne et al. [34] ^b^: CVD ED, lag0–13 respiratory ED visit Martínez-Solanas and Basagaña [31]: 85+ years
	Hyper/hypotension		Lavigne et al. [34] ^b^ Bai et al. [35]		
	Ischemic Heart Disease		Isaksen et al. [33]: percent change in hospitalizations among those 65–85+ years		Isaksen et al. [33]: RR estimate, the percent change in hospitalizations among those 45–64 years
	Dysrhythmia		Bai et al. [35]: compared to minimum morbidity temperature percentile		Bai et al. [35]: compared to the 75th percentile
	Total Stroke	Ha et al. [28]: lag 2	Isaksen et al. [33]: RR estimate, percent change in hospitalizations among those 45–64 and 85+ years Ha et al. [28]: lag2 Martínez-Solanas and Basagaña [31]: 16–64 years, 85+ years		Ha et al. [28]: lag0,1,3 Isaksen et al. [33]: percent change in hospitalizations among those 65–84 years Martínez-Solanas and Basagaña [31]: 75–84 years
	Ischemic Stroke		Ha et al. [28]: lag2–3		Ha et al. [28]: lag0–1
	Hemorrhagic Stroke		Ha et al. [28]		
Heat wave	Total CVD	Chen et al. [29]: heat wave defined by minimum temperature Fuhrmann et al. [41] Guirguis et al. [40] Phung et al. [16]	Chen et al. [29]: heat wave defined by the maximum temperature Sohail et al. [30]: 65–74 (all, short heat waves defined by 90th percentile), 75+ (long heat waves defined by 90th percentile; heat waves defined by 95th percentile)		Sohail et al. [30]: 18–64 (all, short, long heat waves defined by 90th percentile; heat waves defined by 95th percentile), 65–74 (long heat waves defined by 90th percentile; heat waves defined by 95th percentile), 75+ (all, short heat waves defined by 90th percentile)
	Hyper/hypotension	Chen et al. [29]: heat wave defined by the maximum temperature, lag0; heat wave defined by minimum temperature, lag1 Fuhrmann et al. [41]: 2007 heat wave	Chen et al. [29]: heat wave defined by the maximum temperature, lag1; heat wave defined by minimum temperature, lag0 Fuhrmann et al. [41]: 2008 and 2011 heat wave Guirguis et al. [40]		
	Ischemic Heart Disease	Fuhrmann et al. [41]: 2007 and 2008 heat wave Guirguis et al. [40] Chen et al. [29]: heat wave defined by minimum temperature	Fuhrmann et al. [41]: 2011 heat wave Chen et al. [29]: heat wave defined by the maximum temperature, lag0 Sohail et al. [30]: 18–64 (short heat waves defined by 90th percentile), 65–74 (all, short heat waves defined by 90th percentile; heat waves defined by 95th percentile), 75+ (short heat waves defined by 90th percentile; heat waves defined by 95th percentile) Zacharias et al. [42]: ±2.5% excess hospital admissions		Chen et al. [29]: heat wave defined by the maximum temperature, lag1 Sohail et al. [30]: 18–64 (all, long heat waves defined by 90th percentile; heat waves defined by 95th percentile), 65–74 (long heat waves defined by 90th percentile), 75+ (all, long heat waves defined by 90th percentile) Zacharias et al. [42]: ±2.5% excess hospital admissions
	Acute Myocardial Infarction	Guirguis et al. [40] Sohail et al. [30]: 65–74 (short heat waves defined by 90th percentile)	Fuhrmann et al. [41] Sohail et al. [30]: 18–64 (short heat waves defined by 90th percentile), 65–74 (all heat waves defined by 90th percentile; heat waves defined by 95th percentile), 75+ (short heat waves defined by 90th percentile; heat waves defined by 95th percentile)		Sohail et al. [30]: 18–64 (all, long heat waves defined by 90th percentile; heat waves defined by 95th percentile), 65–74 (long heat waves defined by 90th percentile), 75+ (all, long heat waves defined by 90th percentile)
	Dysrhythmia	Guirguis et al. [40] Chen et al. [29]: heat waves defined by minimum temperature, lag0	Chen et al. [29]: heat waves defined by minimum temperature, lag1 Fuhrmann et al. [41] Sohail et al. [30]: 75+ (long heat waves defined by 90th percentile)	Sohail et al. [30]: 18–64 (all heat waves defined by 90th percentile), 75+ (short heat waves defined by 90th percentile)	Chen et al. [29]: heat waves defined by the maximum temperature Sohail et al. [30]: 18–64 (short, long heat waves defined by 90th percentile; heat waves defined by 95th percentile), 65–74 (all, short, long heat waves defined by 90th percentile; heat waves defined by 95th percentile), 75+ (all heat waves defined by 90th percentile; heat waves defined by 95th percentile)
	Heart Failure	Chen et al. [29]: heat waves defined by minimum temperature lag1	Chen et al. [29]: heat waves defined by minimum temperature, lag0 Fuhrmann et al. [41]		Chen et al. [29]: heat waves defined by the maximum temperature
	Total Stroke	Ha et al. [28]: lag2	Ha et al. [28]: lag1,3 Sohail et al. [30]: 18–64 (all, short heat waves defined by 90th percentile), 65–74 (all, short, long heat waves defined by 90th percentile; heat waves defined by 95th percentile), 75+ (all, short, long heat waves defined by 90th percentile; heat waves defined by 95th percentile)	Sohail et al. [30]: 18–64 (heat waves defined by 95th percentile)	Ha et al. [28]: lag0 Sohail et al. [30]: 18–64 (long heat waves defined by 90th percentile)
	Ischemic Stroke	Chen et al. [29]: heat waves defined by the maximum temperature, lag0 Guirguis et al. [40] Ha et al. [28]: lag2,3	Chen et al. [29]: heat waves defined by minimum temperature; heat waves defined by the maximum temperature at lag1 Fuhrmann et al. [41]		Ha et al. [28]: lag0,1
	Hemorrhagic Stroke		Fuhrmann et al. [41]: 2008 Ha et al. [28]: lag0–2		Fuhrmann et al. [41]: 2007, 2011 Ha et al. [28]: lag3

^a^ Examined the association between high temperatures and ED visits for heat stress and dehydration among individuals with pre-existing CVDs. ^b^ Examined the association between high temperatures and ED visits for CVD and respiratory disease among individuals with pre-existing CVDs. Abbreviations: CVD, cardiovascular disease; ED, emergency department; RR, relative risk.

## Data Availability

Not applicable.

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
