# Peer review of "High Temperatures and Cardiovascular-Related Morbidity: A Scoping Review"

_ijerph, 2022, doi:10.3390/ijerph191811243_

Round 1
Reviewer 1 Report
Totally the present article is well-established and the subject is interesting, but some minor revision should be considered.

Reviewer 2 Report
This scoping review is important in understanding the current state of the literature on effects of heat on cardiovascular hospital encounters, from a context relevant to Canada. I have a few comments for the authors' consideration:
-You included reviews if at least half of studies reviewed were conducted in Europe or North America, but some included studies from Asia and Australia. Were there any major differences in reviews that included Asia and Australia compared to studies in Europe or North America?
-I do not see a title for Table 1.
-In the Exposures section under Results, I appreciate your comment that exposures vary between studies. Please comment on the temporal resolutions examined in studies and lags between exposure and outcome. Please also comment on variations in definitions of common terms such as “heat wave” and “heat day.”
-In the discussion, please comment on the applicability of reviewed studies to a Canadian context. Some studies were conducted in Canada, but others were conducted in relatively different climates, such as the southern United States. Also, Canada is a large country with a range of different climates; are there climates in Canada for which you did not have adequate information?
-I think it would be valuable to discuss research gaps and recommended future directions for research in more detail.
Minor comments:
-There seem to be extra bullet points in Figure 1 (top right box and bottom left box).
-Formatting of Table 1 is difficult to read. Either columns should be wider or text should have a smaller font.
-In Table 1, please specify if studied conducted in Washington and New York refer to the city (Washington, DC or New York City) or the state.
-On page 11, second paragraph of “Exposures” section, the font changes.
Reviewer 3 Report
Some content need to be revised or additional clarifications before publication.
1. In the section of 4. Discussion, the direct correlation between high temperatures and CVD in Main Findings can be explained in more detail.
2. It is recommended that the effects of age and gender be described in more detail in the literature.
3. The content in the conclusion did not make it clear what the focus of the results of this paper is. The content should be reorganized so that readers can understand the main points of the paper.
Reviewer 4 Report
Comments
High Temperatures and Cardiovascular-Related Morbidity: A Scoping Review
The subject of the study is relevant in the context of current climate change. Currently many countries are experiencing extreme temperatures that affect the health of the inhabitants.
Much interest has been put into relating these extreme variations to the increase in the incidence of some diseases. People with chronic diseases are very vulnerable to high temperatures since it favors pathophysiological processes that exacerbate their diseases.
Regarding the methodology, it is considered adequate and meets the criteria of a scoping review.
Regarding the results, these reflect the limitations of establishing some relationships or associations that are mostly not significant and this may be due to the heterogeneity of the population and the presence of some biases typical of these studies.
It is recommended in this aspect to consider the biases that may not be declared. Although it is only an exploratory review, it would be advisable to know the possible biases that the selected studies face.
The few existing studies may be due to the impossibility of being able to relate directly (cause - effect) heat with cardiovascular diseases.
It is due to this reason that it is advisable to find methodologies or study designs that are adequate based on this type of study that shows the existing limitations in this regard.
It is recommended to review some words in English since they are misspelled or have typing errors.
Reviewer 5 Report
The article deals with a very interesting topic regarding ‘High Temperatures and Cardiovascular-Related Morbidity: A Scoping Review’. Overall, it is a comprehensive review article and the findings provided indicate that a great deal of effort was put in. Suggestions for improvements that could be performed to the manuscript prior to its publication are the following:
2. Methods:
Line 77: It is suggested to add more relevant references regarding the selected method.
Line 110: Data extraction
A comment regarding the keywords used in the papers that were screened is suggested. Was this piece of information used, and to what extent?
Additional comments about the time distribution of the papers (older or more recent studies) and their findings would be helpful, in order to understand how research on this subject evolved through time.
3. Results:
Line 124 ‘39 were excluded after full text-screening’: Reason? How this could affect the results of the study.
Line 136: The title of Table 1 is missing.
4. Discussion:
Any suggestions about how the authors would confront their study’s limitations and what are the future limitations in case of an extension?
